# Transitional Care Management from Emergency Services to Communities: An Action Research Study

**DOI:** 10.3390/ijerph182212052

**Published:** 2021-11-17

**Authors:** José Batista, Carla Munhoz Pinheiro, Carla Madeira, Pedro Gomes, Óscar Ramos Ferreira, Cristina Lavareda Baixinho

**Affiliations:** 1Vila Franca de Xira Hospital, 2600-009 Vila Franca de Xira, Portugal; carla.madeira@hvfx.min-saude.pt; 2ACES Estuário do Tejo, 2630-242 Arruda dos Vinhos, Portugal; carla.munhoz@arslvt.min-saude.pt; 3Portuguese Institute of Oncology, Nursing Research, Innovation and Development Centre of Lisbon, 1900-160 Lisbon, Portugal; pedro.gomes@campus.esel.pt; 4Nursing School of Lisbon, Nursing Research, Innovation and Development Centre of Lisbon (CIDNUR), 1900-160 Lisbon, Portugal; oferreira@esel.pt (Ó.R.F.); crbaixinho@esel.pt (C.L.B.); 5Center for Innovative Care and Health Technology (ciTechCare), Polytechnic of Leiria, 2411-901 Leiria, Portugal

**Keywords:** aged, caregivers, emergencies, emergency nursing, patient discharge, transitional care

## Abstract

In recent years, nurses have developed projects in the area of hospital to community transition. The objective of the present study was to analyze the transitional care offered to elderly people after they used emergency services and were discharged to return to the community. The action research method was chosen. The participants were nurses, elderly people 70 years old or older, and their caregivers. The study was carried out from October 2018 to August 2019. The data were collected by means of semi-structured interviews with the nurses, analysis of medical records, participatory observation, phone calls to the elderly people and caregivers, and team meetings. The qualitative data were submitted to Bardin’s content analysis. Statistical treatment was carried out by applying SPSS version 23.0. The institution’s research ethics committee approved the research. Only 31.4% of the sample experienced care continuity after discharge, and the rate of readmission to emergency services during the first 30 days after discharge was 33.4%. The referral letters lacked data on information provided to patients or caregivers, and nurses mentioned difficulties in communication between care levels, as well as obstacles to teamwork; they also mentioned that the lack of health policies and clinical rules to formalize transitional care between the hospital and the community perpetuated non-coordination of care between the two contexts. The low level of literacy of patients and their relatives are mentioned as a cause for not understanding the information regarding seeking primary health care services and handing the discharge letter. It was concluded that there is an urgent need to mobilize health teams toward action in the patients’ process of returning home, and this factor must be taken into account in care planning.

## 1. Introduction

The number of people 65 years old or older is expected to increase from 524 million in 2010 to nearly 1.5 billion in 2050 [1], which means there will be changes in health indicators, and this impacts the search for health care. Health-disease transitions and the alterations in functioning resulting from aging, which is made worse by comorbidities, influence elderly people’s quality of life and increase the chances of their becoming dependent on other people [1] to carry out self-care. 

Changes in the epidemiological profile of people who resort to hospital care draw attention to the need to apply different care models to guarantee individualization of care of sick elderly people, as well as of their caregivers. This implies ensuring proper care transition to informal caregivers and deploying social and community resources [2,3] that contribute to care integration, thus preventing care discontinuity, especially after hospital discharge in cases in which patients are dependent [3,4,5,6] and need to have access to health care and support by primary healthcare (PHC) teams guaranteed [5].

In recent years, as a result of studies and recommendations of good practices for transitional care, institutions and professionals have developed some projects to improve training related to informal care and communication between hospitals and different resources in the community [2,3] to solve this complex problem. However, these initiatives are not always supported by health policies and are limited to some inpatient care services. 

There is a consensus that care provided to this population, which shows increased vulnerability, and to its caregivers, still focuses on the solutions of specific problems. It is characterized by being immediate, that is, it does not include delivery of long-term care [4,5,6], especially when patients resort to emergency services (ES) and do not become inpatients. 

Additionally, the problem of overcrowding in ES is made worse by the high number of users who do not have urgent health problems. This reality points to the importance of creating structures and strategies that allow the timely identification of patients who resort to ES with no real need and guarantee the monitoring of these situations by implementing measures targeting adjustment of the access to and the response of different health units [7]. Frequent use of ES by elderly people and overcrowding in these services lead to early discharges and lack of follow-up, which results in increased health costs and reduced care quality [8]. 

Most patients are discharged from ES without receiving proper preparation and, consequently, there are no conditions to prepare the relatives who will play the role of caregivers. Additionally, ongoing support and referral to PHC services are not guaranteed [6,9,10]. Despite these difficulties, this problem can be solved with low-cost actions that provide patients and caregivers with adequate interventions and access to suitable support networks [9,10].

Integrative literature reviews have focused on the hospital transitional care provided to elderly people, and their caregivers have suggested that preparing for this transition benefits elderly people, who often have chronic conditions and are submitted to complex therapies, and their caregivers. These patients are often vulnerable to disarrangements of or failures in care caused by lack of guidance. These reviews have emphasized that this preparation process is still scarce and that few elderly people are currently enjoying its advantages [3,5]. Studies have indicated that evaluations and interventions carried out by nursing teams in ES, with previously trained professionals who have extra competencies regarding transitions between care levels, are effective in decreasing readmission to ES, as well as subsequent complications and loss of function, promoting autonomy and independence for self-care [9,10].

Some authors support the application of a risk evaluation model when elderly people are admitted to ES [8], together with wide-ranging geriatric evaluations. These should be based on a comprehensive approach to providing care to elderly people, with the objective of identifying the characteristics of the population that resorts to ES [11]. This would allow the identification of people at high risk of readmission and timely care planning. Structured interventions, with formal programs for transitions between care levels (including from hospitals to communities), bring gains in the health area: an increase in the perception of quality of life, stabilization of chronic diseases, and reductions in the number of comorbidities. This can result in cost savings resulting from the decreased number of users of ES and shorter hospital stays [8,9,10]. 

In Portugal, there are few studies developed in the area of transitional care between the hospital and the community; the existing ones observe difficulties in the continuity of care between the hospital and the community [2,4,7]. Older people who are admitted to hospital [2,4] or go to the emergency room [7] end up not being followed-up by primary health care.

The objective of the present study was to analyze the transitional care offered to elderly people who sought care in ES and did not experience hospital admission. 

## 2. Materials and Methods

### 2.1. Study Design

Considering the phenomenon in question, the authors opted to use a mixed longitudinal action research method because of its flexibility, which allows the application of a set of interventions that identify and solve clinical practice problems [12,13]. This method also allows interaction between researchers and study subjects, that is, between formal and informal knowledge, and theory and practice, leading to real changes in the way people interact with one another [13]. This encourages actors to cooperate in solving problems [12] and facilitating the transfer of knowledge into clinical practice [13]. 

There are different methodological approaches to this type of study. The authors followed that outlined by Kuhne and Quigley [12], which is organized into three phases and six steps: planning (defining the problem, defining the project, measurement), action (implementing and observing), and reflection (evaluating and verifying if the problem was solved). 

The present study was developed as part of the research project entitled “Transitional care management from emergency services to communities,” which aims to promote transitional care at the hospital–community level offered to vulnerable elderly people and their caregivers.

Given the need to analyze the situation, the following question was established as the starting point of the proposal: “How is transitional care offered to elderly people referred to PHC services carried out after their admission to emergency services?” Based on the recommendations available in the literature, the authors considered that transitional care interventions at the hospital–community level includes three stages: before patients are discharged from hospital, the time of hospital discharge itself, and the period between 48 h and 30 days after hospital discharge [3,5]. 

The present article reports the findings of the planning phase, which occurred from October 2018 to August 2019. In this phase, the health team chose a nurse to act as the project leader; their main activities were communicating with the research team, collecting data, and making the team meetings dynamic. The same professional was supposed to continue the leadership role during the implementation of the intervention and the observation of results, a phase that is currently in progress. The reflection phase, which will focus on the evaluation of the intervention, is expected to occur in the first semester of 2022. 

The study design is represented in Figure 1.

### 2.2. Participants

The present study was carried out in an ES at a Portuguese hospital and its five health centers.

The inclusion criteria for nurses were as follows: providing care directly, agreeing to participate in the study, not being in a situation of absenteeism for over a month during the study period, and participating in the team meetings to diagnose the situation. 

Regarding elderly people, the following eligibility criteria were defined: resorting to the ES without experiencing the need for hospitalization, that is, being discharged from the ES and returning home, being 70 years old or older, having a Barthel index equal to or lower than 55 [14], having Braden scale lower than 16 [2], and having an informal caregiver, identified in the ES. Elderly people who lived in long-term care institutions and those who were between 65 and 70 years old were excluded. This last was decided after the team considered that vulnerability, dependence, and problems associated with aging were more relevant in people at least 70 years old in the context of the present study.

### 2.3. Data Collection

The nature, method, and objective of the study required that data collection be carried out by applying different techniques: semi-structured interviews with the nurses, analysis of medical records, participatory observation, phone calls to the elderly people and caregivers, and team meetings. 

One of the researchers’ conducted semi-structured interviews, with 7 emergency nurses and 16 community nurses, with clinical practice in the follow-up of dependent elderly people in the community. From the question ‘what are the difficulties for continuity of care from the emergency room to the community?’, the experts’ perspective on the subject were explored. The interviews, with an average duration of 25 min, were transcribed.

The interviews were conducted in Portuguese and the whole analysis process, of the different sources, was carried out in this language. A certified translator, with experience in translating qualitative studies, translated the excerpts selected to illustrate the results.

Analysis of medical records focused on two different times: discharge from the ES (to verify if the patient was referred to a PHC service by means of a discharge letter) and 30 days after discharge (to verify the type of care the patients received).

A notebook was used to record field notes, which included data on team meetings, excerpts of the conversations with the nurses, and elements extracted from medical records that allowed the researchers to understand the phenomenon under discussion. These notes were written down by the researcher, who was at the ES during the data collection period. 

Team meetings and times of shift changes were crucial for sharing information, implementing the intervention in everyday procedures, and promoting communication between the researcher and the participants. 

Phone calls to the elderly people and their caregivers allowed the researchers to obtain information regarding difficulties in the hospital–community transition. 

Quantitative and qualitative data were collected simultaneously, and the initial analysis of the first quantitative data guided the qualitative phase and the questions to be asked in interviews and team meetings.

### 2.4. Data Analysis

The data sources were triangulated to obtain the results and ensure in-depth understanding of the phenomenon being studied [12]. Triangulation made it possible to compare and find similarities that increased the validity of the findings.

The analysis of the qualitative findings was performed using computer software (WebQDA^®^ version 2.0, Aveiro, Portugal), which allowed for the organization and analysis of the findings and increased accuracy. The qualitative findings were submitted to Bardin’s content analysis [15]. The coding was performed by the researcher who transcribed the data, which was subsequently validated by the research team. The process of coding and validation of the results involved transcription, reading, coding, definition of categories, and return of the interpretation to the participants for validation of the analysis and interpretation [15].

A code was assigned to each participant and to each source (nurse interview, team meeting, medical records, participatory observation). In the definition of the categories, representativeness, completeness, homogeneity, and relevance were ensured.

Statistical treatment of the quantitative data was carried out using SPSS version 23.0. (USA) The following descriptive statistics parameters were obtained: absolute and relative frequencies, measures of central tendency, and measures of dispersion and variability.

### 2.5. Ethical Considerations

The present study is part of a more comprehensive project concerning safe transitions from hospitals to communities, and the proposed study was approved by the institution’s research ethics committee as per report 09/2019 HVFX. The study complied with the ethical principles laid out in the Declaration of Helsinki: informed consent, privacy, and confidentiality. Because this was an action research study, and researchers were present in the study location, there were concerns regarding the following issues: guaranteeing rigor and objectivity, not imposing values, respecting the participants and their individuality, ensuring fruitful interpersonal relationships that promote well-being, and not putting the participants’ integrity at risk in any situations. 

## 3. Results

### 3.1. Characterization of the Episodes to the Emergency Room

There were 84,068 visits to the general ES between October 2018 and August 2019, of which 9.37% resulted in hospitalization. In 17% of these cases, the patients were discharged without the need for referral, and 3.14% of the patients were referred to insurance companies (occupational and traffic accidents). In 57.6% of the episodes, patients were referred to a PHC service (Table 1).

The number of visits to the ES by people 70 years old or older was 21,178 (25.19%). Of this subset, 4274 patients were discharged from the ES to the community, of whom 226 were referred to a PHC service because of their age group, and also because they had a Barthel index equal to or lower than 55 and a score on the Braden scale lower than 16 points.

The individuals in this realm of elderly people who were referred to a PHC service (*n* = 226) had an average age of 82.7 (±12.3) years and were predominantly women (53.99%). The reasons for visiting the ES differed, but the symptomatology associated with cardiorespiratory (dyspnea, expectoration, hypertension, and cardiothoracic pain) and digestive (vomiting, diarrhea, dehydration) alterations, pain, and falls stood out. 

### 3.2. Continuity of Care after Discharge of the Emergency Room

Examination of the discharge letters drafted at the ES to refer the elderly people to a PHC service showed that all had the following elements: reason for seeking care in the ES, occurrences in the emergency episode, caregiver identification, caregiver phone number, reference health center, family physician, evaluation of Barthel and Braden scales at discharge, and reason for care continuity. Most referral letters lacked data on information provided to patients or caregivers, transmitted knowledge, and indications for accessing PHC services. It is not known whether the absence of these data was a mistake, or if these care items were not carried out. 

Only 71 (31.4%) out of the 226 referred people received continued care in PHC services. There was an average interval of 2.5 days between hospital discharge and the first contact by nurses in the community, with 14.6% of these contacts occurring by phone and 9.7% of the patients being referred to another health unit. In 12.1% of the situations, there was more than one home visit by a nurse (minimum = 1; maximum = 14), and 69.01% (*n* = 49) of the situations required medical appointments. 

Analysis of the medical records at the PHC service showed that the following interventions were developed in the community: evaluating self-care (*n* = 32), evaluating risk of pressure injuries (*n* = 26), managing the environment (*n* = 21), evaluating and assisting with the management of therapeutic regimens (*n* = 21), assisting with patient positioning (*n* = 14), referrals for health services (*n* = 7), providing guidance on inhalation therapy (*n* = 6), evaluating the potential to improve knowledge (*n* = 5), evaluating knowledge to promote health search behaviors (*n* = 5), monitoring vital signs (*n* = 4), and applying dressings to wounds (*n* = 3). 

The rate of readmission to the ES during the first 30 days after hospital discharge was 33.4%, with 21.8% of the readmissions occurring in the first 72 h. Diarrhea and vomiting (17%), pain (headache, sciatica, low back pain) (14.6%), fever (7.3%), vomiting and constipation (4.8%), dyspnea (4.8%), and seizure (4.8%) stood out in the symptomatology that resulted in the visit to the ES. 

### 3.3. Difficulties in Transitional Care

The nurses who participated in the study mentioned, in the interview and in the team meetings, difficulties in transitional care related to health policies, professionals, and patients and their caregivers (Table 2).

#### 3.3.1. Health Policy-Related Difficulties in Transitional Care

During the interviews with the nurses and the meetings held with nurses at the hospital and the PHC service, the participants mentioned that lack of health policies and clinical rules to formalize transitional care between the hospital and the community perpetuated non-coordination of care between the two contexts, despite international recommendations and professionals’ awareness of the advantages involved: 

‘… *quite often we wonder why the debate gets back to the ES, why there was no intervention in the health center, but the reality is that the population sees the ES as the gateway to the hospital. This is an old issue that has not been solved, and the lack of investment in health centers, the lack of action there, contribute to the problem*’.(Nurse 7)

‘*There must be legislation for care continuity to work. A good example is the coordination of the discharge management teams and the coordination with the network units. It is defined, there is a platform for information communication*’.(Nurse 13)

‘*The Health Data Platform allows access to some data in the medical record, diagnoses, tests, admission, and discharge dates, but for the clinical practice of nurses it is poor; the focuses and needs do not get recorded*’.(Nurse 5)

This lack of formalization of transitional care makes it difficult for institutions to assign care hours that allow nurses to plan elderly people’s discharge based on the level of urgency and promote coordination with PHC services: 

‘… *while it is not mandatory and a priority for services, it is impossible to do it; there is no time and no direct gains for the institution, because the funding is based on the number of emergency episodes*’.(Team meeting 1)

‘*Improving care continuity implies having time to send e-mails to your colleagues or directly calling to refer patients and transmit information*’.(Team meeting 3)

#### 3.3.2. Professional-Related Difficulties in Transitional Care 

The nurses mentioned that there were difficulties involving communication between care levels, as well as obstacles to teamwork:

‘*In many cases, patients are discharged from the ES; the doctor discharges them and they leave the services without contact with the nursing team. That is why we cannot identify all who need care continuity*’.(Nurse 2)

‘*We send the discharge letter, which we know does not always get to the health center, but we do not know who it should be addressed to, which professional ensures care to that patient and their family (…) Patients themselves do not know who the nurses who pay home visits are*’.(Nurse 25)

‘*When we receive discharge letters from the hospital, we evaluate the level of urgency of the situation, but these patients often need medical care and we have only one doctor, who has one afternoon a week to go to the patients’ houses to evaluate and prescribe*’.(Nurse 24)

#### 3.3.3. Patient and Relative-Related Difficulties in Transitional Care 

Nurses at both the hospital and the PHC service pointed out that patients did not know how to access either community care teams or social and community resources. The low level of literacy of patients and their relatives was also recognized as a cause for not understanding the information regarding seeking PHC services and handing the discharge letter:

‘*I was contacted by a caregiver who was asking for a wheelchair for her mother. I know that family well and I found it strange, because the elderly woman attended her appointments to check on her diabetes with no gait difficulties. During the conversation, I realized that three weeks before C had fallen when getting out of bed to go to the day care center, gone to the hospital ES, and been discharged with the indication to rest. Her daughter did not notice this indication and left her mother in bed for three weeks (…) The letter had information about getting up and resting over specific periods, alternatingly*’. (Nurse 23)

## 4. Discussion

The results of the present study have reinforced the moral, economic, and human imperative of guaranteeing transitional care, and complex health-disease transitions, to dependent people. These requirements are both organizational and developmental, because this population requires support and specific skills in order to be successful and so the care initiated in the hospital can be maintained [3,16]. In situations where patients have cognitive alterations and/or show moderate to total dependence, caregivers take responsibility for their care, replacing patients in the roles of carrying out self-care, managing therapeutic regimens, and getting access to health care. However, for that to happen, they must receive support in their transition into the role of caregiver [5].

When discharge from ES occurs, not all elderly people and relatives have been prepared to return home or been referred to a PHC service, and those who have been referred by means of a discharge letter often end up not having their right to care continuity guaranteed. This problem was reported in another study that pointed out that transitional care is neglectful, including many independent interventions carried out by nurses, such as helping patients with getting up, therapeutic positioning, personal hygiene care, and health education [17]. A number of factors can contribute to care omissions: the high number of interventions that require continuity over long periods, the time necessary to properly plan the process of returning home, communication difficulties, lack of evaluation of intervention effectiveness, and lack of recording and systematization of the protocols used [3,16,17,18]. 

Difficulties related to communication and coordination between care levels are real and hinder integrated responses to the needs of the population with complex health-disease problems [16,18]. This has led researchers to suggest that criteria for care continuity be defined. A study with orthopedic patients recommended that criteria be defined for care continuity oriented toward both patients and caregivers, in order to direct clinical decisions and interventions in transitional care [2]. In the part concerning patients, there is a focus on criteria related to the risk of ineffective management of therapeutic regimens, such as compromised patient mobility, high risk of fall and/or pressure injury, management of pain and therapeutic regimen, and risk of nonadherence to rehabilitation programs. Regarding relatives, the recommended criteria are related to the transition into the role of caregiver and continuity of care, which demands certain levels of knowledge and competence from caregivers [2,19]. 

It is also important to emphasize the need to improve communication networks and empower patients and their families by providing them with knowledge and adaptation strategies in the face of the requirements and risks involved in returning home. An integrative literature review that had the objective of identifying the needs of caregivers of elderly people in the transition from hospital to home presented five categories: needs in the transition into the role of caregiver, self-care needs, health needs, economic needs, and social and community needs. It is noteworthy that three subcategories emerged in the first category: information, skills to carry out care, and emotional support and overload prevention [5].

The high patient turnover rate in ES and the internal organization of this setting, which allows patients to go directly from a physician’s office into the streets without interacting with nurses, may hinder team communication and coordination in the ES itself. Considering that health-disease transitions bring about dependence and do not allow immediate functional recovery to pre-hospitalization levels, and that this care omission is a cause of readmissions, it is necessary to implement measures that make patients’ adaptation to their new situation of depending on their family feasible in order to guarantee self-care once these patients return home [3,5,16,18]. 

The high rate of return to ES during the first 72 h after discharge found in the present study reinforced the fact that, when elderly patients return home, some care items cannot be postponed or not carried out, at the risk of worsening the symptomatology and a return to ES. Early visits by community teams allow therapy continuity and also help identify new needs of patients and family caregivers, resulting in responses aligned with the problems detected, promotion of collaboration between care teams, symptom management, increased care and education offered to family caregivers, and coordination of social and health services. Future studies must explore coordination between ES and communities.

A study in France estimated that patients 75 years old or older had a rehospitalization rate over the first 30 days after hospital discharge of 14%, and it considered that one-quarter of these episodes were preventable [20]. To control the risk of recurrent visits to ES, the authors highlighted the need for transitional care interventions, which should be maintained for at least 30 days after the visit to the ES or hospitalization [20,21,22]. Interventions that bridge the gap between hospital and home and involve professionals assigned for this purpose (usually nurses) are more effective in reducing the risk of rehospitalization [20], and they are planned and patient-centered, ensuring care continuity and anticipating control of possible complications [21]. 

Nursing interventions must provide for the training of patients so they can carry out activities of daily living [23], the better integration of caregivers in the discharge process [24], and more formalized support for medication management [22,23]. Additionally, these interventions must guarantee access to useful contacts and guidance regarding what to do when there are signs of decompensation or worsening of disease. They should also provide educational programs regarding self-management [25] and support to caregivers [26].

Nurses refer to the absence of national policies that reinforce the need to ensure continuity of care between the hospital and the community, although this integration of care remains a challenge. Guaranteeing this coordination between care levels (considering the risks in different phases of the cycle) and ensuring transitional care is imperative for developing a more sustainable health system [27]. The World Health Organization has been warning about this problem since 2016 and has recommended policies to consolidate patient-centered health services and reorganize the care model, with an emphasis on transitional care interventions based on advanced practice nursing as a priority with an impact on the decrease hospital costs and readmission rates [28].

### Limitations

The study has limitations associated with the action research method. The data obtained in a very specific context do not allow for replication, especially qualitative findings. The triangulation of various sources increased the validity of the study and provides a more holistic picture of the study.

The findings of this study are limited to the study context and, therefore, are not generalizable. However, these limitations regarding the rigor with which the research was conducted allows the transferability of the results to similar contexts and allows the discussion of this phenomenon.

## 5. Conclusions

The ageing process, which is associated with the high prevalence of chronic diseases and accidents, leads to increased demand for emergency services by older people and their caregivers. This is a challenge for health services organization, especially regarding the process of returning home, as well as for communication and transition between care levels. The need to adapt the response of health systems so they can fight the possible care fragmentation is indisputable, as this risk increases the likelihood of readmittance to the emergency room, inadequate follow-up, and lack of preparation and information of the caregivers, and it is a predictor of difficulties in adhering to the therapeutic regime, difficulties in accessing health care, worsening of symptoms, and functional decline and loss of life quality.

In spite of the above, there are a limited amount of studies on transitional care, from hospital to community, with older persons that have an episode of emergency, without need of hospitalization. The literature review showed that there are difficulties in ensuring continuity of care between these two levels of care, which is reinforced by the results of this study; although a discharge letter is sent, it does not always reach the primary health care professionals, which indicates the need to improve communication between these two levels of care. In most situations, the person leaves the emergency department with moderate to severe dependence and at home do not have the support of the primary health care professionals, returning to the emergency room within the first 72 h after discharge.

Another issue is that patients and caregivers do not participate in the planning of hospital discharge and decision-making when they have to return home. Health professionals mentioned lack of time to offer this care modality as a difficulty.

The nurses’ intervention in the community is focused on the assessment of self-care needs, assessment of the risk of pressure ulcers, management of the environment, assessment and assistance in the management of the therapeutic regimen, and positioning. 

The findings of this research reinforce the urgency to mobilize nursing teams for action in anticipating the return home, which should be a central concern in care planning before the person leaves the emergency department to ensure safety, avoid breaks in the continuity of care, and as an effective measure to prevent readmissions in the immediate post-discharge period. The existence of a member of the health team (nurse) with expertise in care transition, who is exclusively responsible for assessing the person, planning discharge, and referring him/her to community resources in order to ensure timely interventions to promote coping skills, disease, and symptom management, and caregiver empowerment can have a significant impact on preventing defragmentation in the continuity of care and reducing visits to the emergency department.

National and local health policies are needed to strengthen transitional care and increase professionals’ adherence to referral, ensuring continuity of care and the support of people in their health-disease transitions.

We conclude by stating that coordination and integration of care between hospital settings and community teams remains a challenge and shows a tendency to reduce complications, health resources consumption and home follow-up, and, consequently, hospital bed occupation rates. It is necessary to guarantee training, follow-up, and coordination between care levels. 

## Figures and Tables

**Figure 1 ijerph-18-12052-f001:**
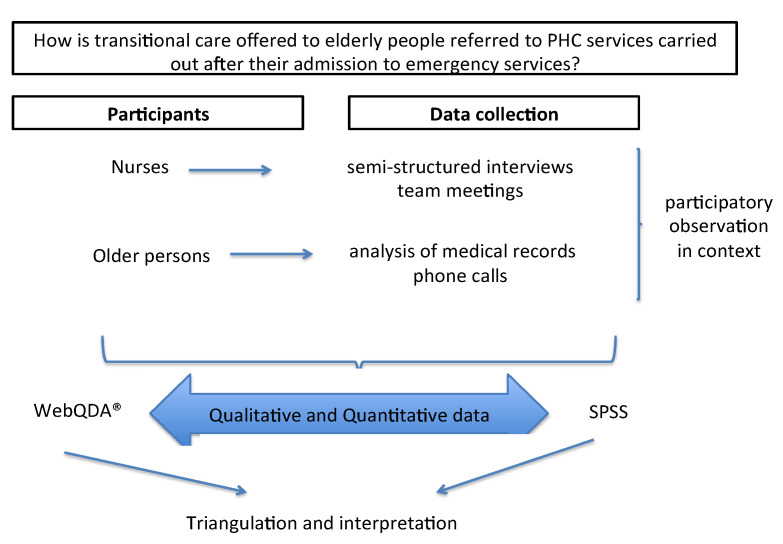
Study design. Vila Franca de Xira; Portugal.

**Table 1 ijerph-18-12052-t001:** Characterization of the visits to the ES. Vila Franca de Xira, Portugal. 1 October 2018 to 31 August 2019.

Step Following the Visit to the ES	*n*	%
Withdrawal (during medical care)	914	1.08
Disciplinary discharge	2	--
Referral to a PHC service	48,283	57.6
External appointment	9040	10.8
External (insurance company)	2637	3.14
External (not mentioned)	11,675	13.8
Death followed by autopsy	37	0.04
Death not followed by autopsy	128	0.16
Hospital in the area where the patient lived	121	0.15
Hospital that did not belong to the Portuguese National Health System	10	0.01
Hospital that belonged to the Portuguese National Health System	1146	1.36
Departure against medical advice	363	0.44
No medical care after screening	1830	2.11
Inpatient service	7882	9.31
Total	84,068	100

**Table 2 ijerph-18-12052-t002:** Difficulties in transitional care; Portugal. 1 October 2018 to 31 August 2019.

Category	Subcategory	Registration Units
Difficulties in transitional care	Health policy-related difficulties in transitional care	*n* = 71
	Professional-related difficulties in transitional care	*n* = 84
	Patient and relative-related difficulties in transitional care	*n* = 47

## Data Availability

The data used during this study are available from the corresponding author, under request by e-mail.

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
