# Peer review of "Transitional Care Management from Emergency Services to Communities: An Action Research Study"

_ijerph, 2021, doi:10.3390/ijerph182212052_

Round 1

Reviewer 1 Report

Thank you for sending your paper entitled “Transitional care management from emergency services to communities: an action research study.” to International Journal of Environmental Research and Public Health. After carefully review this interesting paper, the following comments are listed for your reference:

To increase potential citations, authors should check keywords against those recommended in the MESH Browser of Medical Subject Headings https://meshb.nlm.nih.gov/search
1.    Abstract (P1, L16-29): It would be desirable to include when the study was carried out in the abstract section. In addition, I would suggest adding more qualitative information in the results. On the other hand, the sentence “the results of the present study cannot be generalized” is part of the limitation of the study, so I would recommend removing it from the abstract and including it in limitations.
2.    Keywords (P1, L30): Keywords should be listed alphabetically.
3.    Introduction (P3, L86): Is there a mistake in the abbreviation “US”. Should it be “ES”?
4.    Introduction (P2, L92-93): I would suggest a connector in the last sentence before including the aim of the study to keep the reading flowing. 
5.    Methods (P3-4, L143-159): I would recommend including more information in the Data collection section such as how many semi-structured interviews were conducted, whether they were audio recorded, how long they lasted, the interview protocol used, etc. 
6.    Methods (P4, L161-166): I would recommend including more information in the Data analysis section such us the software used for qualitative data; the steps followed in the content analysis to determine how the data was coded.
7.    Methods: It might be a good idea to include a quantitative and qualitative phase to provide more details.
8.    Methods: Were the interviews conducted in English? If not, I would suggest explaining how the translation process was carried out in this section.
9.    Results: It might be a good idea to include a qualitative and qualitative phase section in the results to differentiate them for the readers' understanding, and it would also be desirable to provide more details. 
10.    Results (P5, L224-278): For a better and easier reading experience, I would suggest clearly separating paragraphs (perhaps by putting quotes in cursive). Furthermore, I would recommend more space between paragraphs on different themes or categories.
11.    Limitations (P8, L353): I would recommend including a limitations section at the end of the discussion and providing more details in it.
12.    Conclusions (P8, L355-370): The first paragraph of the conclusions are your results. I would suggest rewriting this section.

Author Response

Dear review:

First of all thank you for taking the time to review our article and for the recommendations that improved the overall quality of the article.

We attach the response to the amendments requested, indicating the amendment and its location in the text.

Reviewer 2 Report

Congratulations on the project and the manuscript. The subject is very interesting and the appropriate approach. The mixed methodology is very relevant. In general the manuscript is too theoretical. The methodology is explained in detail but a poor discussion is offered. The conclusions also contain results, which is not relevant. I recommend readjusting the text in general: reducing the introduction, specifying and reorganizing the methodological part and improving the conclusions. Finally, I believe that the statistical analysis carried out is very poor and I recommend carrying out bivariate analyzes.

Other comments :

Introduction: nothing to object. Well written and the objective is clear.

Methodology: the chosen research design is complex but well described from a theoretical point of view. However, I suggest the inclusion of a figure that visually explains the design and that a specific objective is added in the introduction about this part of the project. On the other hand, I consider that the inclusion and exclusion criteria are clear but I have doubts about the data collection. What mechanism has been used to control the precision of the data? I think triangulation needs to be better explained. Regarding the data analysis, the quantitative part is extremely poor and it is surprising that the SPSS is mentioned only for frequencies and percentages. Qualitative analysis also needs to be explained (only mentions Bardin’s content analysis), how has it been categorized? In general, I think that this section needs to be rewritten since it has a lot of filler information and it has important details that have been lost.  

Results: I suggest a table with the categories and subcategories of the qualitative analysis  

Discussion: they need to delve into the limitations of this study to be able to generalize the results with the necessary caution.  

Conclusions: they are very brief and not very relevant. They should not contain results.  

Author Response

Dear review:

We thank you for the careful review of our manuscript which allowed us to improve its overall quality.
We attach the response to the requests.

Best regards;

Round 2

Reviewer 1 Report

Thank you for sending your revised paper entitled “Transitional care management from emergency services to communities: an action research study” to IJERPH and take my recommendations into account.

However, some recommendations were not addressed and may need to be explained within the text:

  1. Methods: Were the interviews conducted in English? If not, I would suggest explaining how the translation process was carried out in this section.
  2. Conclusions (P10, L422-428): Your results are still mentioned in the first paragraph of the conclusions. This section should be rewritten with proper conclusions.

Author Response

Dear review:

Once again thank you for taking the time to review our article.

With regard to the question:

  1. Methods: Were the interviews conducted in English? If not, I would suggest explaining how the translation process was carried out in this section.

We introduced in page 4 (lines 168-170) the follow information - The interviews were conducted in Portuguese and the whole analysis process, of the different sources, was carried out in this language. A certified translator, with experience in translating qualitative studies, translated the excerpts selected to illustrate the results.

In relation to the comment:

2. Conclusions (P10, L422-428): Your results are still mentioned in the first paragraph of the conclusions. This section should be rewritten with proper conclusions.

We changed the conclusions page 10 - lines 426-447; 462-469.

Best Regards;

Reviewer 2 Report

Thank you very much for the changes made. I think the manuscript is more balanced and better described. My general perception of the text has improved a lot, especially because of the improvements made to the methodology and conclusions. Thank you very much for considering my criteria and congratulations on the study.

Author Response

Dear review:

We thank you once again for your time and attention in reviewing our article.
We send it with some changes shaded in green (page 4 - lines 168-170; page 10 - lines 426-447 and 462 - 469).

Best regards;